# Differential Effects of Fkbp4 and Fkbp5 on Regulation of the *Proopiomelanocortin* Gene in Murine AtT-20 Corticotroph Cells

**DOI:** 10.3390/ijms22115724

**Published:** 2021-05-27

**Authors:** Kazunori Kageyama, Yasumasa Iwasaki, Yutaka Watanuki, Kanako Niioka, Makoto Daimon

**Affiliations:** 1Department of Endocrinology and Metabolism, Hirosaki University Graduate School of Medicine, 5 Zaifu-cho, Hirosaki, Aomori 036-8562, Japan; y_wtnk@hirosaki-u.ac.jp (Y.W.); kniioka@hirosaki-u.ac.jp (K.N.); mdaimon@hirosaki-u.ac.jp (M.D.); 2Department of Clinical Nutrition Management Nutrition Course, Faculty of Health Science, Suzuka University of Medical Science, 1001-1 Kishioka-cho, Suzuka, Mie 510-0293, Japan; iwasakiyasumasa@gmail.com

**Keywords:** adrenocorticotropic hormone, proopiomelanocortin, pituitary, Fkbp, glucocorticoid

## Abstract

The hypothalamic-pituitary-adrenal axis is stimulated in response to stress. When activated, it is suppressed by the negative feedback effect of glucocorticoids. Glucocorticoids directly inhibit *proopiomelanocortin* (*Pomc*) gene expression in the pituitary. Glucocorticoid signaling is mediated via glucocorticoid receptors, 11β-hydroxysteroid dehydrogenases, and the FK506-binding immunophilins, Fkbp4 and Fkbp5. Fkbp4 and Fkbp5 differentially regulate dynein interaction and nuclear translocation of the glucocorticoid receptor, resulting in modulation of the glucocorticoid action. Here, we explored the regulation of *Fkbp4* and *Fkbp5* genes and their proteins with dexamethasone, a major synthetic glucocorticoid drug, in murine AtT-20 corticotroph cells. To elucidate further roles of Fkbp4 and Fkbp5, we examined their effects on *Pomc* mRNA levels in corticotroph cells. Dexamethasone decreased *Pomc* mRNA levels as well as *Fkpb4* mRNA levels in mouse corticotroph cells. Dexamethasone tended to decrease Fkbp4 protein levels, while it increased *Fkpb5* mRNA and its protein levels. The dexamethasone-induced decreases in *Pomc* mRNA levels were partially canceled by *Fkbp4* knockdown. Alternatively, *Pomc* mRNA levels were further decreased by *Fkbp5* knockdown. Thus, Fkbp4 contributes to the negative feedback of glucocorticoids, and Fkbp5 reduces the efficiency of the glucocorticoid effect on *Pomc* gene expression in pituitary corticotroph cells.

## 1. Introduction

Stress activates the hypothalamic-pituitary-adrenal (HPA) axis. The activated HPA axis is suppressed by the negative feedback effect of glucocorticoids. Corticotropin-releasing factor (CRF), produced in the hypothalamic paraventricular nucleus in response to stress, stimulates the release of the adrenocorticotropic hormone (ACTH) from the anterior pituitary [1,2,3]. ACTH, cleaved from the *proopiomelanocortin* (*Pomc*) gene, stimulates the secretion of corticosterone and cortisol, the principal glucocorticoid in rodents and human, respectively, from the adrenal glands [3]. Glucocorticoids bind to the glucocorticoid receptor (GR), and subsequently inhibit CRF production in the hypothalamus and the ACTH in the pituitary as an inhibitory feedback loop [4,5,6]. 

Glucocorticoid signaling is mediated via the GR, 11β-hydroxysteroid dehydrogenases, and the FK506-binding immunophilins, Fkbp52 (Fkbp4) and Fkbp51 (Fkbp5) [7,8,9]. The genes encoding Fkbp4 and Fkbp5 are *Fkbp4* and *Fkbp5*, respectively. Fkbp4 and Fkbp5 differentially regulate dynein interaction and nuclear translocation of the GR [10]. In the absence of corticosterone, the GR is retained in the cytoplasm as a complex containing one GR molecule, heat shock protein (HSP) 90 dimer, HSP90-binding protein P23, and Fkbp5 [11,12]. 

Corticosterone binds to a cytoplasmic receptor, GR, in rodents. Following the binding of corticosterone to the GR, Fkbp5 is replaced by Fkbp4, resulting in translocation of the complex to the nucleus owing to the interaction between Fkbp4 and the motor protein dynein [12]. Fkbp5 changes the conformation of the receptor complex, leading to sensitivity reduction of the GR to corticosterone and negative feedback efficiency [13]. 

We revealed regulation and roles of Fkbp4 and Fkbp5 in corticotroph cells. In the present study, we explored the regulation of *Fkbp4* and *Fkbp5* genes and their proteins using dexamethasone, a major synthetic glucocorticoid drug, in murine corticotroph AtT-20 cells. To elucidate further roles of Fkbp4 and Fkbp5, we subsequently examined the effects of Fkbp4 and Fkbp5 on *Pomc* mRNA levels in corticotroph cells. 

## 2. Results

### 2.1. Effect of Dexamethasone on Pomc mRNA Levels 

This time course study showed that 100 nM dexamethasone significantly decreased *Pomc* mRNA levels with marked effects observed within the first 24 h of treatment (*p* < 0.05, Figure 1A). *Pomc* mRNA levels decreased in a concentration-dependent manner (*p <* 0.005), with significant effects initially occurring at 1 nM dexamethasone (Figure 1B).

### 2.2. Effect of Dexamethasone on Fkbp4 mRNA Levels 

This time course study showed that 100 nM dexamethasone significantly decreased *Fkbp4* mRNA levels (*p* < 0.01). Within the first 24 h of incubation with dexamethasone, *Fkbp4* mRNA levels decreased to 72% of the control level (Figure 2A). Additionally, *Fkbp4* mRNA levels decreased as dexamethasone concentrations increased (*p* < 0.01), with significant effects initially occurring at 10 nM dexamethasone (Figure 2B). 

### 2.3. Effect of Dexamethasone on Fkbp5 mRNA Lvels 

This time course study showed that 100 nM dexamethasone significantly increased *Fkbp5* mRNA levels (*p* < 0.0001). Within the first 6 h of dexamethasone incubation, *Fkbp5* mRNA levels increased to 375% of the control level (Figure 3A). Additionally, *Fkbp5* mRNA levels were increased as dexamethasone concentrations increased (*p* < 0.0001), with significant effects initially occurring at 1 nM dexamethasone (Figure 3B).

### 2.4. Effect of Dexamethasone on Fkbp4, Fkbp5, and GR Protein Levels

This time course study showed that 100 nM dexamethasone significantly increased Fkbp5 (*p* < 0.005) and tended to decrease Fkbp4 protein levels (*p* = 0.094) (Figure 4A and 4B). Within the first 24 h of dexamethasone incubation, Fkbp5 protein levels were increased to 192% of the control level (Figure 4B), while GR protein levels were decreased to 47% (Figure 4C).

### 2.5. Effect of Fkbp4 and Fkbp5 Knockdown on Dexamethasone-Induced Changes of Pomc mRNA Levels 

*Fkbp4* mRNA levels were reduced by 18% in cells transfected with si*Fkbp4*, while *Fkbp5* mRNA levels remained unchanged. *Pomc* mRNA levels were not changed in cells transfected with si*Fkbp4*, while they were decreased by dexamethasone. Dexamethasone-induced decreases in *Pomc* mRNA levels were partially canceled by *Fkbp4* knockdown (Figure 5A). 

*Fkbp5* mRNA levels were reduced by 55% in cells transfected with si*Fkbp5*. As expected, *Fkbp4* mRNA levels were not affected by *Fkbp5* knockdown. *Pomc* mRNA levels were not changed in cells transfected with si*Fkbp5*, while they were decreased by dexamethasone (Figure 5B). The dexamethasone-induced decreases in *Pomc* mRNA levels were further decreased by *Fkbp5* knockdown (Figure 5B).

## 3. Discussion

The activated HPA axis is suppressed by the negative feedback effect of glucocorticoids [14]. Pituitary corticotroph cells express high levels of GRs, and glucocorticoids directly inhibit *Pomc* gene expression in pituitary corticotroph cells [15]. However, the molecular mechanisms for glucocorticoid negative regulation of *Pomc* gene expression are not fully understood. Glucocorticoid suppression of the *Pomc* gene is specific for the pituitary, while *Pomc* gene expression is upregulated by glucocorticoids in the hypothalamus [16]. King et al. [17] proposed that the differences of transcription factors among cells or tissues cause differential regulation, because the interactions with specific DNA-regulatory sequences and other transcription factors produce cell-type specific effects. 

By binding glucocorticoid to the GR, the GR translocates from the cytoplasm to the nucleus [12]. The glucocorticoid-GR complex directly and indirectly regulates target gene transcription. The glucocorticoid-GR complex binds to glucocorticoid-response elements (GRE) in the target gene promoter, and subsequently activates target gene transcription [18], while the negative glucocorticoid-response elements (nGREs) are necessary for the negative regulation of *Pomc* gene expression by glucocorticoids [4,19]. The GR complex binds to nGREs of the *Pomc* promoter, and the nGRE complex suppresses *Pomc* transcription in the pituitary corticotroph [12]. Additionally, suppression of Nur77 or NeuroD1 by glucocorticoids is also involved in the glucocorticoid-mediated negative regulation of *Pomc* in the pituitary [16,20]. 

In our study, dexamethasone-induced decreases in *Pomc* mRNA levels were partially canceled by *Fkbp4* knockdown. Following the binding of corticosterone to the GR, Fkbp5 is replaced by Fkbp4, resulting in translocation of the complex to the nucleus (Figure 6A). Newly-formed GR/HSP90/Fkbp4 complexes generally accumulate in the nucleus [21]. Thereafter, the GR acts on the expression of target genes. Thus, Fkbp4 contributes to the negative feedback effect of glucocorticoids. *Fkbp4* mRNA levels decreased 24 h after incubation with dexamethasone, and Fkbp4 protein levels tended to decrease. To induce homeostasis, the decrease in Fkbp4 could lead to a decrease in the effects on *Pome* gene expression of glucocorticoids.

GR activity directly stimulates *Fkbp5* gene transcription, and upregulates the protein FKPB5 [22]. Thereafter, Fkbp5 inhibits GR activity [10]. In our study, dexamethasone increased *Fkpb5* mRNA and Fkbp5 protein levels. *Fkbp5* gene expression was reported to be induced by glucocorticoids in humans and rodents [23,24,25]. In fact, the human *Fkbp* gene contains numerous sites for GR binding [26]. Glucocorticoids activate the *Fkbp5* gene and increase the protein content, resulting in inhibition of GR translocation to the nucleus (Figure 6B). Our present study shows that dexamethasone-induced decreases in *Pomc* mRNA levels were further decreased by *Fkbp5* knockdown. Thus, the increase in Fkbp5 also diminishes the effects on *Pome* gene expression of glucocorticoids. Additionally, in cooperation with the decrease in GR, Fkbp5 may contribute to the desensitization of glucocorticoid action after the long-term exposure.

Other molecules, such as HSP90, HSP90-binding protein P23, and GR itself, may be involved in these glucocorticoid effects. Post-tanslational regulation of these molecules, such as protein phosphorylation, acetylation, and SUMOylation, as well as expression regulation, may also contribute to the effects. For example, HSP90 hyperacetylation results in a loss of chaperone activity [27]. Riebold et al. [28] also showed that Cushing’s disease is caused by overexpression of HSP90 protein and can be treated with an appropriate HSP90 inhibitor. It remains undetermined whether dexamethasone also affects HSP90 expression or its activity in pituitary corticotroph cells. 

In this study, mouse corticotroph tumor cells were used. However, it is unclear whether these results would be obtained in normal corticotroph cells. In fact, corticotroph tumor cells may show glucocorticoid resistance compared with normal corticotroph cells. Therefore, it is possible that the expression levels of Fkbp4 and Fkbp5 might differ between these cell types. Studies to determine this are required in future. 

In conclusion, dexamethasone decreased *Pomc* mRNA levels as well as *Fkpb4* mRNA levels in the mouse corticotroph. Dexamethasone increased *Fkpb5* mRNA and its protein levels. Moreover, dexamethasone-induced decreases in *Pomc* mRNA levels were partially canceled by *Fkbp4* knockdown, and were further decreased by *Fkbp5* knockdown. Thus, Fkbp4 contributes to the negative feedback of glucocorticoids, and Fkbp5 reduces the efficiency of the glucocorticoid effect on *Pomc* gene expression in pituitary corticotroph cells.

## 4. Materials and Methods

### 4.1. Materials

Dexamethasone was purchased from Sigma-Aldrich (St. Lois, MO, USA). 

### 4.2. Cell Culture

Murine pituitary AtT-20 corticotroph tumor cells were obtained from ATCC (Manassas, VA, USA). The cells were cultured in Dulbecco’s modified Eagle’s medium (DMEM) (Sigma-Aldrich) supplemented with 10% fetal bovine serum (FBS), 100 µg/mL streptomycin, and 100 U/mL penicillin at 37 °C in a humidified atmosphere (5% CO_2_ and 95% air). Cells were seeded in six-well plates at 15.0 × 10^4^ cells/cm^2^ for 2 day prior to each experiment. On day 3, cells were washed with DMEM supplemented with 0.2% bovine serum albumin (BSA), and subsequently cultured overnight in DMEM without FBS prior to each experiment. Total cellular RNA was collected at the conclusion of each experiment, and stored at –80 °C until required for the relevant assay.

### 4.3. RNA Extraction

Cells were incubated at the indicated times with medium alone (control) or medium containing 100 nM dexamethasone. To examine the concentration-dependent effects of dexamethasone, cells were incubated with medium alone (control) or medium containing 1–100 nM dexamethasone. Total cellular RNA was extracted using the RNeasy Mini Kit (QIAGEN, Hilden, Germany) according to the manufacturer’s protocol. The extracted RNA (0.5 µg) was subjected to a real-time (RT) reaction using random hexamers as primers with the SuperScript First-Strand Synthesis System for the quantitative RT polymerase chain reaction (RT-qPCR) (Thermo Fisher Scientific, Waltham, MA, USA) as described previously [29].

### 4.4. RT-qPCR

The resulting cDNA was subjected to RT-qPCR as follows. mRNA expression levels of mouse *Pomc* and *Fkbp4/5* were evaluated using RT-qPCR with transcript-specific primer and probe sets (Assays-on-Demand Gene Expression Products; Applied Biosystems, Foster City, CA, USA). To standardize gene expression levels, β2-microglobulin (*B2mg*) was used as a reference gene. Across all treated samples, *B2mg* mRNA levels did not significantly differ from those of controls. 

The 25-µL RT-qPCR reactions consisted of 1 × TaqMan Universal PCR Master Mix (Applied Biosystems) and 1 × Assays-on-Demand Gene Expression Products for each of the transcripts (Mm00435874_m1 for mouse *Pomc*, Mm00487391_m1 for mouse *Fkbp4*, Mm00487401_m1 for mouse *Fkbp5*, and Mm00487401_m1 for mouse *B2mg*), and 500 ng cDNA (25 µL total volume). An ABI PRISM 7000 Sequence Detection System (Applied Biosystems) was used for amplification with the following thermal cycling conditions: 95 °C for 10 min followed by 40 cycles at 95 °C for 15 s and 60 °C for 1 min. All data are expressed as a function of the threshold cycle (C_T_) for quantitative analyses using ABI PRISM 7000 SDS software (Applied Biosystems). Analyses that used diluted samples of the gene of interest and the reference gene (*B2mg*) revealed identical amplification efficiencies. Relative quantitative gene expression was calculated using the 2^−ΔΔC^_T_ method.

### 4.5. RNA Interference Experiments 

*Fkbp4/5* and control siRNA fragments were designed and purchased from QIAGEN. The HiPerFect transfection reagent (QIAGEN) was used to transfect AtT-20 cells with siRNA fragments according to the manufacturer’s protocol. 

Target mRNA levels in samples were determined from cells that were seeded in 6-well plates at a density of 15 × 10^4^ cells/well. Cultures were incubated for 24 h in 1 mL of culture medium control or experimental siRNAs: *Fkbp4*-specific siRNA (si*Fkbp4*, Mm_Fkbp4_1) or *Fkbp5*-specific siRNA (si*Fkbp5*, Mm_Fkbp5_8), and subsequently incubated in BSA medium containing dexamethasone or control for 24 h. Thereafter, *Pomc*, *Fkbp4, Fkbp5*, and *B2mg* transcript levels were assayed via qRT-PCR.

### 4.6. Western Blot Analysis 

Western blot analysis was performed to examine the effect of dexamethasone (100 nM) on changes in the protein expression of Fkbp4, Fkbp5, and GR. β-actin was used as a housekeeping protein. Cells were washed twice with phosphate-buffered saline (PBS; Life Technologies, Grand Island, NY) and lysed with Laemmli sample buffer. Cell debris was pelleted via centrifugation and the supernatant was recovered. Samples were boiled and subjected to electrophoresis on a 4–20% gradient polyacrylamide gel, and proteins were transferred to a polyvinylidene fluoride membrane (Daiichi Kagaku, Tokyo, Japan). After blocking with Detector Block^®^ buffer (Kirkegaard & Perry Laboratories, Gaithersburg, MD), the membrane was incubated at room temperature for 1 h with each antibody (anti-Fkbp4 antibody [1:5000 dilution]; anti-Fkbp5 [1:2500 dilution] antibody; anti-GR antibody [1:5000 dilution] (Proteintech Group, Rosemont, IL); and anti-β-actin [1:20,000 dilution] antibody, ab8227, Abcam, Cambridge, MA), washed with PBS containing 0.05% Tween 20, and incubated with horseradish peroxidase-labeled anti-rabbit immunoglobulin G (1:20,000 dilution; Daiichi Kagaku). The chemiluminescent substrate SuperSignal West Pico (Pierce Chemical Co., Rockford, IL) was used for detection, and the membrane was exposed to BioMax film (Eastman Kodak Co., Rochester, NY).

### 4.7. Statistical Analyses

Each in vitro experiment was performed three times. Samples were analyzed in triplicate for each group of experiments. Data are expressed as means ± standard errors of the means. Analysis of variance was performed, followed by Shceffe’s multiple comparison tests. Results with *p* values < 0.05 were considered significant.

## Figures and Tables

**Figure 1 ijms-22-05724-f001:**
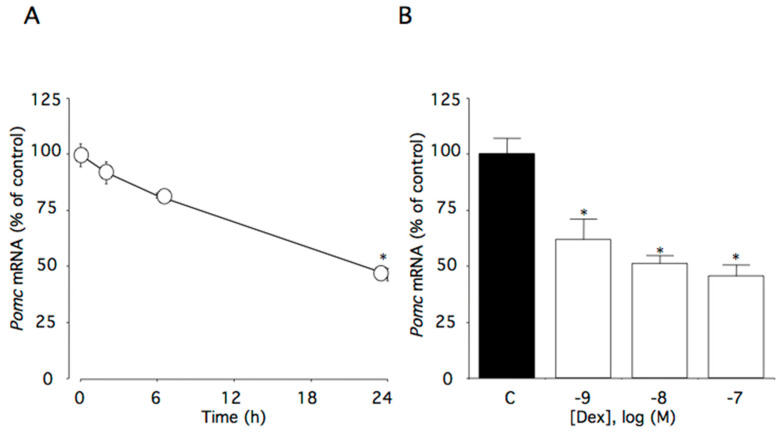
Effects of dexamethasone on *Pomc* mRNA levels in AtT-20 cells. (**A**) Time-dependent effects of dexamethasone on *Pomc* mRNA levels. Cells were cultured in medium containing 100 nM dexamethasone. (**B**) Concentration-dependent effects of dexamethasone on *Pomc* mRNA levels. Cells were cultured for 24 h in medium containing 1–100 nM dexamethasone. Data are expressed as means ± standard errors of the means. * *p* < 0.05 compared with time 0 or the control (C). Cells were treated in triplicate, and the average of three independent experiments is shown (n = 3).

**Figure 2 ijms-22-05724-f002:**
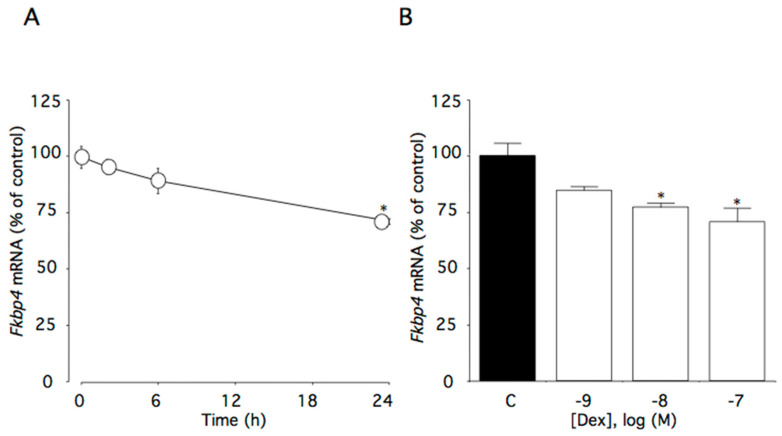
Effects of dexamethasone on *Fkbp4* mRNA levels in AtT-20 cells. (**A**) Time-dependent effects of dexamethasone on *Fkbp4* mRNA levels. Cells were cultured in medium containing 100 nM dexamethasone. (**B**) Concentration-dependent effects of dexamethasone on *Fkbp4* mRNA levels. Cells were cultured for 24 h in medium containing 1–100 nM dexamethasone. Data are expressed as means ± standard errors of the means. * *p* < 0.05 compared with time 0 or the control (C). Cells were treated in triplicate, and the average of three independent experiments is shown (n = 3).

**Figure 3 ijms-22-05724-f003:**
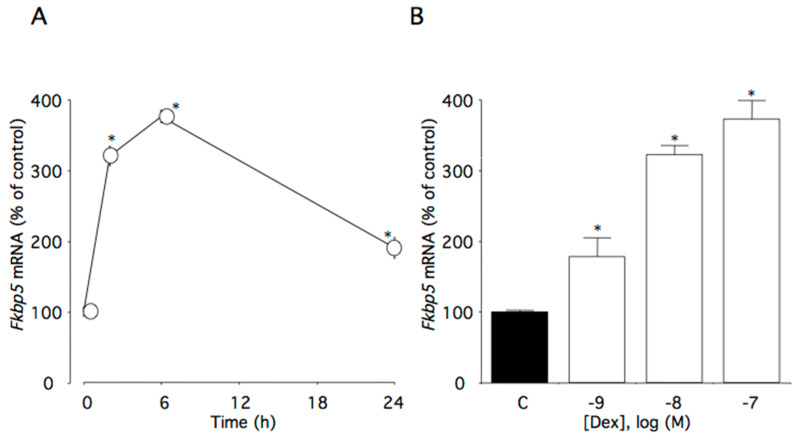
Effects of dexamethasone on *Fkbp5* mRNA levels in AtT-20 cells. (**A**) Time-dependent effects of dexamethasone on *Fkbp5* mRNA levels. Cells were cultured in medium containing 100 nM dexamethasone. (**B**) Concentration-dependent effects of dexamethasone on *Fkbp5* mRNA levels. Cells were cultured for 6 h in medium containing 1–100 nM dexamethasone. Data are expressed as means ± standard errors of the means. * *p* < 0.05 compared with time 0 or the control (C). Cells were treated in triplicate, and the average of three independent experiments is shown (n = 3).

**Figure 4 ijms-22-05724-f004:**
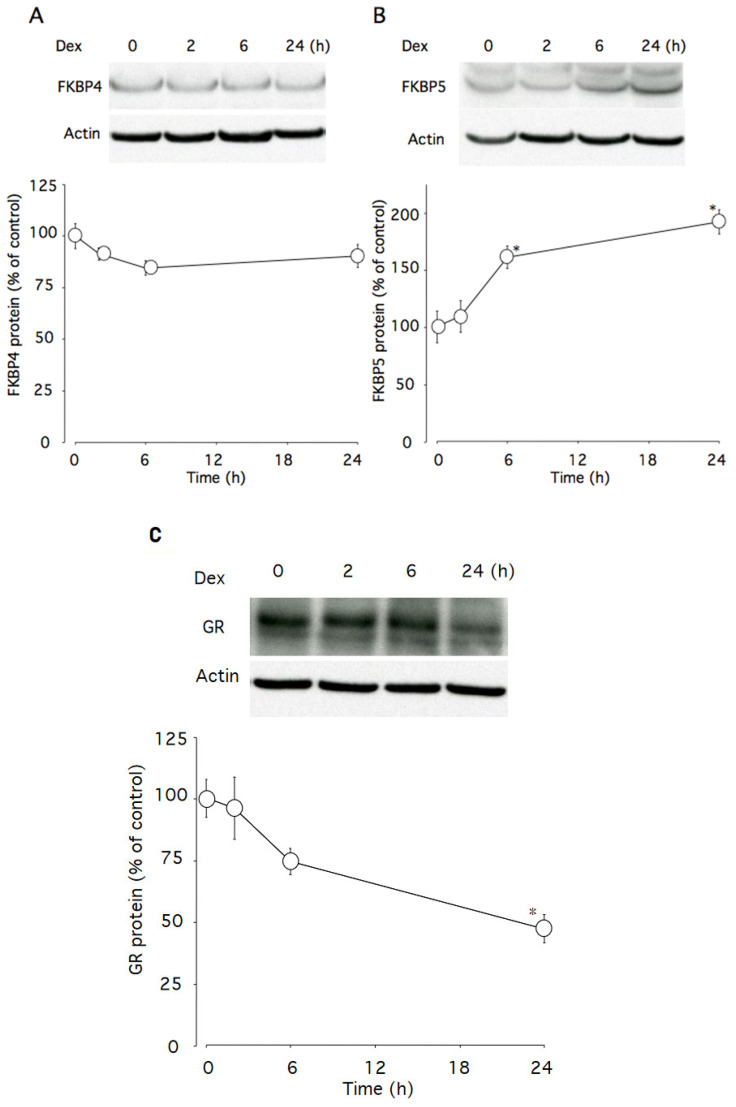
Effects of dexamethasone on Fkbp4 and Fkbp5 proteins levels in AtT-20 cells. (**A**) Time-dependent effects of dexamethasone on Fkbp4 protein levels. (**B**) Time-dependent effects of dexamethasone on Fkbp5 protein levels. (**C**) Time-dependent effects of dexamethasone on GR protein levels. Cells were cultured in medium containing 100 nM dexamethasone. β-actin was used as a housekeeping protein. Data are expressed as means ± standard errors of the means. * *p* < 0.05 compared with time 0. Cells were treated in triplicate, and the average of three independent experiments (n = 3) and a representative blot are shown.

**Figure 5 ijms-22-05724-f005:**
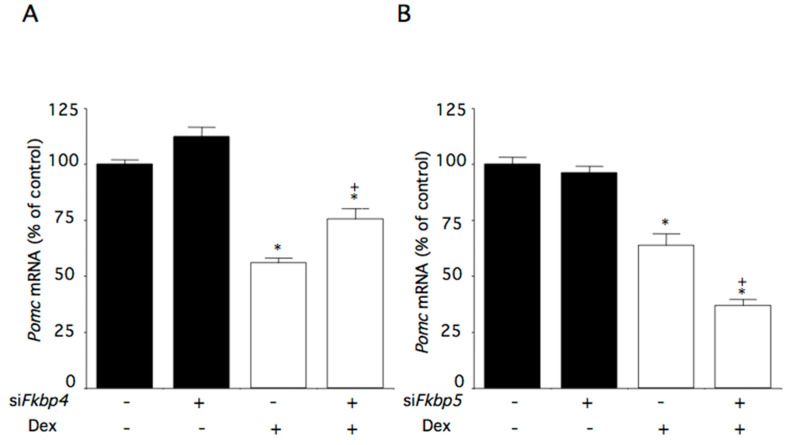
Effects of Fkbp4 and Fkbp5 knockdown on dexamethasone-induced changes of *Pomc* mRNA levels in AtT-20 cells. Cells were incubated with medium containing control small interfering (si)RNA or *Fkbp4-* or *Fkbp5*-specific siRNA (si*Fkbp4* or *siFkbp5*), and subsequently with medium containing 100 nM dexamethasone (Dex) or control medium. (**A**) Effect of Fkbp4 on dexamethasone-induced changes of *Pomc* mRNA levels. (**B**) Effect of Fkbp5 on dexamethasone-induced changes of *Pomc* mRNA levels. Data are expressed as means ± standard errors of the means. * *p* < 0.05 compared with control siRNA and Dex (-). ^+^
*p* < 0.05 compared with control *siFkbp* and the Dex (-) group or control siRNA and the Dex (+) group. Cells were treated in triplicate, and the average of three independent experiments is shown (n = 3).

**Figure 6 ijms-22-05724-f006:**
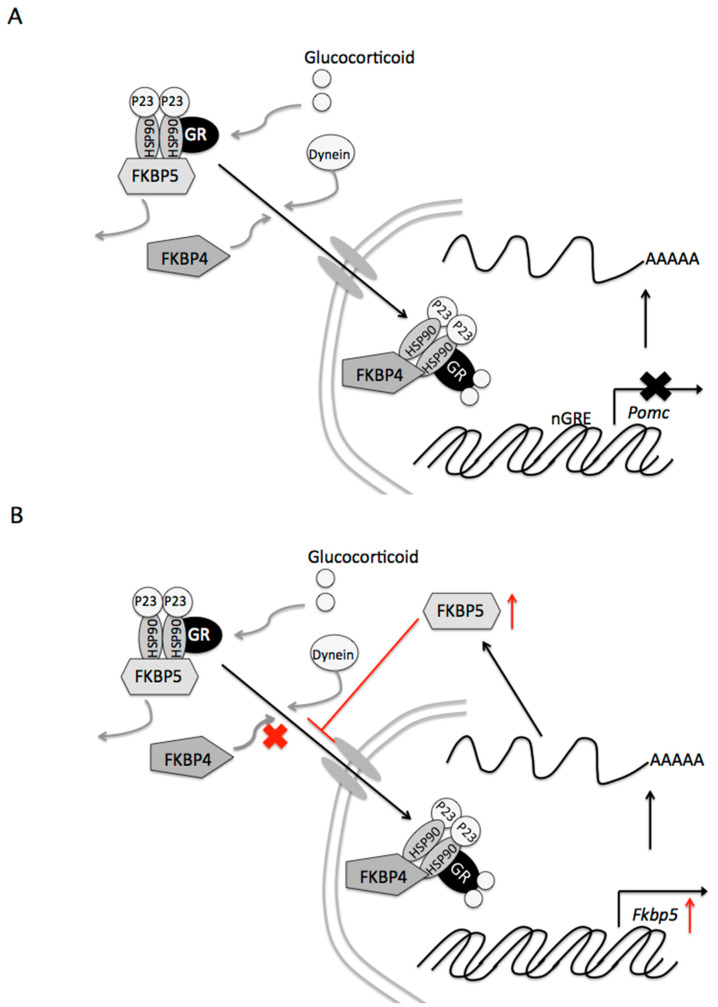
Proposed signaling mechanisms of Fkbp4 and Fkbp5 by glucocorticoids in pituitary corticotroph cells. (**A**) Contribution of Fkbp4 to the negative feedback of glucocorticoids. Following binding of glucocorticoid to the glucocorticoid receptor (GR), Fkbp5 is replaced by Fkbp4, resulting in translocation of the complex to the nucleus. Newly-formed GR/heat shock protein 90 (HSP90)/Fkbp4 complexes generally accumulate in the nucleus. The GR subsequently suppresses the expression of the target *Pomc* gene. (**B**) Reduced efficiency of the glucocorticoid effect of Fkbp5 on *Pomc* gene expression. GR activity stimulates *Fkbp5* gene transcription and upregulates the protein FKPB5 (indicated by a red arrow). Increases in the Fkbp5 protein inhibit GR translocation to the nucleus, resulting in diminished effects on target genes of glucocorticoids.

## Data Availability

Data are contained within the article.

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
