# Peer review of "Differential Effects of Fkbp4 and Fkbp5 on Regulation of the Proopiomelanocortin Gene in Murine AtT-20 Corticotroph Cells"

_ijms, 2021, doi:10.3390/ijms22115724_

Round 1
Reviewer 1 Report
The present study Kazunori Kageyama et al examined the changes of Fkbp4 and Fkbp5 genes and their respective protein levels on dexamethasone treatment in murine AtT-20 corticotroph cells. They showed that dexamethasone decreased Pomc mRNA and Fkpb4 mRNA levels only, but increased both mRNA and protein levels of FKPB5. Furthermore, dexamethasone-induced decreases in Pomc mRNA levels were partially canceled by Fkbp4 knockdown, but were further decreased by Fkbp5 knockdown. The author concluded that Fkbp4 contributes to the negative feedback of glucocorticoids, while Fkbp5 reduces the efficiency of the glucocorticoid effect on Pomc gene expression in pituitary corticotroph cells. Overall, this is a potentially interesting study trying to elucidate the underlying mechanisms of FKBP4 and FKBP5 in dexamethasone feedback. Some of the results are novel and well performed. However, the study for now stays in correlational changes and thus are severely compromised as listed the following: 1.The results that Fkbp4 knockdown partially blocked dexamethasone-induced Pomc mRNA decreases do not validate Fkbp4 contributes to the negative feedback of glucocorticoids. Pomc mRNA levels can be the direct result of Fkbp4 knockdown as indicated previously. Thus, more experiments are required to examine the relations between Pomc Fkbp4 mRNA levels and dexamethasone treatment. 2.Similarly, Fkbp5 knockdown decreased dexamethasone-induced Pomc mRNA changes do not necessarily suggest that Fkbp5 reduces the efficiency of the glucocorticoid effect. Other genetic manipulations like Fkbp5 overexpression are needed to arrive at this conclusion. 3.Glucocorticoids level and GR protein expression changes should be provided after dexamethasone treatment. 4.The protein expression changes of Fkbp4 should also be compared, as the mRNA expression level of Fkbp5 at the 6th and 24th hour time points are not parallel to the protein expression level(figure3、4). 5.The mRNA levels of Fkbp4/5 after transfected with siFkbp4/5 in cells need to be presented.Author Response
Response to Reviewers
We would like to thank the reviewers and the editor for their careful review of the manuscript and constructive comments.
We have addressed all of the criticisms made by the reviewers and have modified our manuscript according to the suggestions. Our manuscript was revised on the basis of the reviewers’ comments and critiques (changes indicated with yellow highlight).
We still believe that this paper will be of interest to the readership of the journal because it provides insight into the molecular biology of the proopiomelanocortin gene, which is in line with the aims and scope of the esteemed journal.
We hope that the revised manuscript is now acceptable for publication in IJMS.
The revised manuscript was proofread by a native speaker.
Reviewer 1:
The present study Kazunori Kageyama et al examined the changes of Fkbp4 and Fkbp5 genes and their respective protein levels on dexamethasone treatment in murine AtT-20 corticotroph cells. They showed that dexamethasone decreased Pomc mRNA and Fkpb4 mRNA levels only, but increased both mRNA and protein levels of FKPB5. Furthermore, dexamethasone-induced decreases in Pomc mRNA levels were partially canceled by Fkbp4 knockdown, but were further decreased by Fkbp5 knockdown. The author concluded that Fkbp4 contributes to the negative feedback of glucocorticoids, while Fkbp5 reduces the efficiency of the glucocorticoid effect on Pomc gene expression in pituitary corticotroph cells. Overall, this is a potentially interesting study trying to elucidate the underlying mechanisms of FKBP4 and FKBP5 in dexamethasone feedback. Some of the results are novel and well performed. However, the study for now stays in correlational changes and thus are severely compromised as listed the following:
1.The results that Fkbp4 knockdown partially blocked dexamethasone-induced Pomc mRNA decreases do not validate Fkbp4 contributes to the negative feedback of glucocorticoids. Pomc mRNA levels can be the direct result of Fkbp4 knockdown as indicated previously. Thus, more experiments are required to examine the relations between Pomc Fkbp4 mRNA levels and dexamethasone treatment.
Response: We greatly appreciate your review and comments. Our responses to the comments are shown in the following section. We sincerely appreciate all of your constructive suggestions regarding our paper.
We agree with your opinion that additional experiments would strengthen our results. However, the overexpression constructs of FKBP4 were currently not available. Furthermore, overexpression experiments require a considerable amount of time and are particularly complex. In our view, gene knock-down produces results that are more physiologically relevant than overexpression since overexpressed cells generally gain artificial levels of the target gene. Although we would like to conduct overexpression studies in the future, we request permission to bypass these experiments in the current manuscript.
2.Similarly, Fkbp5 knockdown decreased dexamethasone-induced Pomc mRNA changes do not necessarily suggest that Fkbp5 reduces the efficiency of the glucocorticoid effect. Other genetic manipulations like Fkbp5 overexpression are needed to arrive at this conclusion.
Response: We appreciate your suggestion and agree with your opinion. However, similarly, the overexpression constructs of FKBP5 were also currently not available. In addition, we consider that gene knock-down produces results that are more physiologically relevant than overexpression.
3.Glucocorticoids level and GR protein expression changes should be provided after dexamethasone treatment.
Response: According to your suggestion, GR protein expression changes were provided after dexamethasone treatment. However, glucocorticoid levels after dexamethasone treatment were unable to be determined by the current assay system.
Page 4, the last line: GR protein levels were decreased to 47% (Figure 4C).
Page 8, 4th paragraph, the last line: Additionally, in cooperation with the decrease in GR, FKBP5 may contribute to the desensitization of glucocorticoid action after the long-term exposure.
4.The protein expression changes of Fkbp4 should also be compared, as the mRNA expression level of Fkbp5 at the 6th and 24th hour time points are not parallel to the protein expression level (figure3、4).
Response: According to your suggestion, the protein expression changes of Fkbp4 were also compared.
Page 8, line 197: Fkbp4 mRNA levels decreased 24 h after incubation with dexamethasone, and FKBP4 protein levels tended to decrease. To induce homeostasis, the decrease in FKBP4 could lead to a decrease in the effects on Pome gene expression of glucocorticoids.
5.The mRNA levels of Fkbp4/5 after transfected with siFkbp4/5 in cells need to be presented.
Response: The mRNA levels of Fkbp4/5 after transfection with siFkbp4/5 in cells were presented in the Results.
Page 6, Results 2.5.: Fkbp4 mRNA levels were reduced by 18% in cells transfected with siFkbp4. Fkbp5 mRNA levels were reduced by 55% in cells transfected with siFkbp5.

Reviewer 2 Report
The authors have shown the involvement of Fkbp4 and Fkbp5 in the negative feedback mechanism of POMC gene expression by glucocorticoids. The experimental system is simple and the results are very easy to understand. Reviewers suggest adding the following considerations to make this treatise more advanced:
Much of the involvement of Fkbp4 and Fkbp5 in the expression of glucocorticoid action has already been clarified. Since the authors use an experimental system using ACTH-producing tumor cell lines, reviewers require consideration of differences in expression regulatory function between normal and tumors.
The translocation of glucocorticoid receptors into the nucleus involves many molecules, as the authors show in Fig. 6. Reviewers require a review of the literature as to whether other molecules, including Hsp90, have a glucocorticoid-induced expression regulation mechanism. Reviewers also ask for explanations about the involvement of post-translational regulation, such as protein phosphorylation, acetylation, and SUMOylation, as well as expression regulation.
Author Response
Response to Reviewers
We would like to thank the reviewers and the editor for their careful review of the manuscript and constructive comments.
We have addressed all of the criticisms made by the reviewers and have modified our manuscript according to the suggestions. Our manuscript was revised on the basis of the reviewers’ comments and critiques (changes indicated with yellow highlight).
We still believe that this paper will be of interest to the readership of the journal because it provides insight into the molecular biology of the proopiomelanocortin gene, which is in line with the aims and scope of the esteemed journal.
We hope that the revised manuscript is now acceptable for publication in IJMS.
The revised manuscript was proofread by a native speaker.
Reviewer 2:
The authors have shown the involvement of Fkbp4 and Fkbp5 in the negative feedback mechanism of POMC gene expression by glucocorticoids. The experimental system is simple and the results are very easy to understand. Reviewers suggest adding the following considerations to make this treatise more advanced:
Much of the involvement of Fkbp4 and Fkbp5 in the expression of glucocorticoid action has already been clarified. Since the authors use an experimental system using ACTH-producing tumor cell lines, reviewers require consideration of differences in expression regulatory function between normal and tumors.
Response: We appreciate the reviewer’s favorable comment. According to the suggestion, we addressed the differences in expression regulatory function between normal and tumor cells.
Page 8, the last paragraph: In this study, mouse corticotroph tumor cells were used. However, it is unclear whether these results would be obtained in normal corticotroph cells. In fact, corticotroph tumor cells may show glucocorticoid resistance compared with normal corticotroph cells. Therefore, it is possible that the expression levels of FKBP4 and FKBP5 might differ between these cell types.
The translocation of glucocorticoid receptors into the nucleus involves many molecules, as the authors show in Fig. 6. Reviewers require a review of the literature as to whether other molecules, including Hsp90, have a glucocorticoid-induced expression regulation mechanism. Reviewers also ask for explanations about the involvement of post-translational regulation, such as protein phosphorylation, acetylation, and SUMOylation, as well as expression regulation.
Response: We appreciate the reviewer’s suggestions. In accordance, we addressed other molecules and post-translational regulation.
Page 8, 5th paragraph: Other molecules, such as HSP90, HSP90-binding protein P23, and GR itself, may be involved in these glucocorticoid effects. Post-tanslational regulation of these molecules, such as protein phosphorylation, acetylation, and SUMOylation, as well as expression regulation, may also contribute to the effects. For example, HSP90 hyperacetylation results in a loss of chaperone activity [27]. Riebold et al. [28] also showed that Cushing’s disease is caused by overexpression of HSP90 protein and can be treated with an appropriate HSP90 inhibitor. It remains undetermined whether dexamethasone also affects HSP90 expression or its activity in pituitary corticotroph cells.

Round 2
Reviewer 1 Report
the authors have addressed all my concerns and I have no further questions.